# Nexus between PI3K/AKT and Estrogen Receptor Signaling in Breast Cancer

**DOI:** 10.3390/cancers13030369

**Published:** 2021-01-20

**Authors:** Aditi S. Khatpe, Adedeji K. Adebayo, Christopher A. Herodotou, Brijesh Kumar, Harikrishna Nakshatri

**Affiliations:** 1Department of Surgery, Indiana University School of Medicine, Indianapolis, IN 46202, USA; askhatpe@iu.edu (A.S.K.); akadebay@iu.edu (A.K.A.); cherodot@iu.edu (C.A.H.); kumarbr@iupui.edu (B.K.); 2Department of Biochemistry and Molecular Biology, Indiana University School of Medicine, Indianapolis, IN 46202, USA; 3VA Roudebush Medical Center, Indiana University School of Medicine, Indianapolis, IN 46202, USA

**Keywords:** breast cancer, estrogen receptor, PI3K-AKT-mTOR, anti-estrogen resistance

## Abstract

**Simple Summary:**

Breast cancers are broadly classified into two subtypes: estrogen receptor-positive and estrogen receptor-negative. Approximately 70% of breast cancers are estrogen receptor-positive and this type of breast cancer is more common in postmenopausal women. Estrogen receptor-positive breast cancers are treated with a class of drugs called anti-estrogens. While the majority of tumors respond to this class of drugs, disease recurs in approximately 30% of cases, sometimes even 20 years after initial diagnosis. This review highlights efforts to understand why tumors recur despite effective treatments and outcome of these efforts in the development of new combination therapies. At least three new types of combination therapies that delay progression of recurrent tumors are in clinical use.

**Abstract:**

Signaling from estrogen receptor alpha (ERα) and its ligand estradiol (E2) is critical for growth of ≈70% of breast cancers. Therefore, several drugs that inhibit ERα functions have been in clinical use for decades and new classes of anti-estrogens are continuously being developed. Although a significant number of ERα+ breast cancers respond to anti-estrogen therapy, ≈30% of these breast cancers recur, sometimes even after 20 years of initial diagnosis. Mechanism of resistance to anti-estrogens is one of the intensely studied disciplines in breast cancer. Several mechanisms have been proposed including mutations in *ESR1*, crosstalk between growth factor and ERα signaling, and interplay between cell cycle machinery and ERα signaling. *ESR1* mutations as well as crosstalk with other signaling networks lead to ligand independent activation of ERα thus rendering anti-estrogens ineffective, particularly when treatment involved anti-estrogens that do not degrade ERα. As a result of these studies, several therapies that combine anti-estrogens that degrade ERα with PI3K/AKT/mTOR inhibitors targeting growth factor signaling or CDK4/6 inhibitors targeting cell cycle machinery are used clinically to treat recurrent ERα+ breast cancers. In this review, we discuss the nexus between ERα-PI3K/AKT/mTOR pathways and how understanding of this nexus has helped to develop combination therapies.

## 1. Introduction

Breast cancer is one of the leading causes of death in women. According to the American Cancer Society report of 2019, 79% of total breast cancer cases are estrogen receptor alpha (ERα)-positive. The five-year survival rate of women with ERα+ breast cancer is around 90% [1]. With recent advancements, survival rate of breast cancer patients has improved significantly. However, a major challenge is the development of resistance to available therapies. For example, previously responsive ERα+ tumors show eventual resistance to the anti-estrogen tamoxifen [2]. Therefore, the study of resistance mechanisms to endocrine therapy requires an understanding of ER structure, molecular pathways, and interaction with components of other signaling cascades.

## 2. Biology of ERs

Estrogen receptors belong to the nuclear receptor superfamily [3]. The two different forms of ER—ERα and ERβ—are coded by two distinct genes *ESR1* and *ESR2*, which are located on chromosomes 6 and 14, respectively [4]. ERs are composed of six functional domains, similar to other members of the nuclear receptor family [3]. The N-terminal A/B domain bears the ligand independent activation function 1 (AF1) to which various transcription coregulators and activators bind. The DNA binding domain (DBD), which is also referred as C domain, is important for interaction of ER with the genome. The DBD of ER binds to cis-regulatory element termed estrogen response element (ERE) and activates estrogen responsive genes such as pS2/TFF1, GREB1, and IGFBP4 [5]. Other domains include the D-domain, also known as the hinge domain, which contains nuclear localization signal; E domain or ligand binding domain (LBD) to which ligands bind and the nonconserved F domain [6,7]. The DBD of ERα and ERβ are 97% identical, whereas the LBD shares 56% homology. Hence, individual ligand–receptor interactions activate distinct pathways through recruitment of different coactivator/corepressor molecules, thereby, altering the transcriptional profile. Crystal structures have revealed that the DBD-LBD organization forms a distinct L-shaped boot structure [8]. This spatial organization has been reported to be critical for receptor function. This structure can be perturbed by small molecules and formed the basis for developing many clinically used anti-estrogens [8].

ERs are randomly distributed in the cell and are maintained in an inactive state by the corepressor proteins including NCoR family of proteins such as NCoR1, SMRT [9]. At least 19 proteins with corepressor activity have been described and these corepressors recruit histone deacetylases (HDACs) to mediate the actions of anti-estrogens [9]. To activate ER-dependent transcription, ER-corepressor complexes need to be dissociated and replaced by coactivator complexes. More than 400 coactivators have been described in the literature and several of them can bind to ERα in a ligand-dependent manner [10]. As estrogen diffuses through cell membrane, ER encounters the ligand and binding occurs. This binding leads to conformational changes in ER and dissociation of ER-corepressor complex. The dissociation of inhibitory proteins activates the receptor, which then leads to homo- or heterodimerization. ER can also undergo such conformational changes through post-translational modifications including phosphorylation (described below). Phosphorylated receptor dimers are then transported into the nucleus for the transcription of ER target genes. Activated ER binds to ERE and recruits coregulatory molecules to initiate transcription. Binding of ER to chromatin and subsequent activation of gene expression is additionally controlled by a distinct group of transcription factors called pioneer factors [11]. Although there is still some debate on whether pioneer factors facilitate ER binding or ER facilitates pioneer factor binding, we recently reported chromatin accessibility changes in estradiol treated cells and observed a modest but significant enrichment of pioneer factor binding sites in gene regulatory regions of genes induced by E2 suggesting that ER facilitates pioneer factor binding in at least a subset of genes [12]. Since many excellent reviews on the relationship between pioneer factors and ERs have been published [11,13], ER-pioneer factor interactions are not discussed further.

Estrogens regulate activity of ERs by serving as ligands. Estrone (E1), Estradiol (E2), and Estriol (E3) are three major forms of estrogens. E2 is the most important ligand of ERs. Androgens are converted into estrogens by the enzyme aromatase through a process called aromatization. Synthesis of estrogens takes place mainly in the ovaries and the reaction is stimulated by follicle-stimulating hormone (FSH) and luteinizing hormone (LH). Therefore, prophylactic oophorectomy reduces the risk of recurrence and is advisable in many breast cancer patients [14,15].

As described above, the majority of ERα-E2 actions is within in the nucleus. However, depending on the cell type, a small fraction of ERα is involved in nongenomic action by tethering to the plasma membrane [16]. Plasma membrane bound ERα may interact with growth factor receptors upon E2 binding, which subsequently leads to activation of nonreceptor tyrosine kinases such as SRC. This in turn leads to phosphorylation of aromatase, increase in aromatase activity and a resulting de novo synthesis of E2. As a consequence, an aromatase-E2 autocrine feedforward loop gets activated with an integration of genomic and nongenomic actions of ERα [17] (Figure 1).

Varieties of transcription-independent signal transduction pathways are activated through nongenomic actions of ERα through SRC. Nuclear and membrane ERα exist in 9:1 ratio, although ratio varies between cell types [18]. This 10% of membrane ERα manipulates the transcription profile of the cell. Modifications such as palmitoylation (Cys447) and myristoylation aid in tethering of ERα to the plasma membrane [19,20]. Similar to nuclear ERα, membrane bound ERα exists mainly as homodimers [21]. The E2-activated membrane ERα undergoes depalmitoylation, dissociates from the membrane, and then interacts with signaling molecules such as PI3K [22]. Consequently, signals downstream of PI3K, including AKT are activated, which promote cell proliferation and survival. Other signaling pathways activated by the membrane ERα is the Mitogen-activated protein kinase (MAPK/Erk1/2) [23,24,25]. In cancer, activation of this cascade leads to tumor growth and progression. Figure 1 provides a summary of multiple mechanisms of ERα activation and actions.

## 3. PI3K-AKT-mTOR Signaling Axis in Breast Cancer

AKT, also referred to as protein kinase B (PKB), is part of the critical PI3K-AKT-mTOR pathway deregulated in multiple cancers [26]. There are three functional AKTs encoded by distinct genes, AKT1, AKT2, and AKT3 (also called PKBα, PKBβ, and PKBγ) [27,28]. The AKT3 isoform can be alternatively spliced, resulting in different expression and activation patterns, which further complicates expression/activity landscape of AKTs [29]. Structurally, AKTs consist of a central serine/threonine kinase domain, an N-terminal Pleckstrin Homology (PH) domain, and a hydrophobic C-terminal tail. The major pathway of AKT activation is through the class I phosphatidylinositol 3-kinase (PI3K) family [28]. Receptor tyrosine kinases (RTKs) engage extracellular growth factors and respond by activating PI3K at the cell membrane. PI3K converts phosphatidylinositol 4,5-bisphosphate (PIP2) to phosphatidylinositol (3,4,5)-triphosphate (PIP3), which interacts with PH domain of AKT and transfers AKTs to the cell membrane along with phosphoinositide-dependent protein kinase 1 (PDK1). This confers conformational changes in AKTs that expose T308 and S473, which are phosphorylation sites in the kinase-domain and the regulatory C-terminal domains of AKTs, respectively [30,31]. At the membrane, AKT is phosphorylated by PDK at T308 in a PIP3-dependent manner. Further PIP3-dependent phosphorylation by mTORC2 at S473 is required for full AKT activation [28]. Facchinetti and colleagues described mTORC2-dependent phosphorylation of T450, which is responsible for AKT folding and stability [32]. 

The tumor suppressor protein PTEN is a negative regulator of AKT as it converts PIP3 back to PIP2, limiting the duration of activation of the cascade. Predictably, inactivating *PTEN* mutations have been shown to be oncogenic mutations and important risk factors of breast cancer [33]. Other mechanisms also contribute to AKT activation. For example, EGF activation of AKT in breast cancer is mediated by calmodulin [34]. Furthermore, earlier studies have demonstrated the importance of GTP-bound Ras-GAP in the activation of PI3K downstream of platelet-derived growth factor (PDGF) signaling, which highlights crosstalk between PI3K-AKT-mTOR and Ras-Raf-MEK-ERK pathways [35]. Further demonstrating this interconnectedness is a report suggesting that AKT upregulates macrophage inhibitory cytokine-1 (MIC-1) expression, which in turn increases activation of ERK1 [36]. Additional pathways that regulate AKTs include mRNA methylation of upstream AKT regulators, aberration of normal miRNA control over AKT and its regulators, changes in ubiquitination of the PH domain, causing failure to localize to the membrane, and changes in regulation by lncRNAs [26].

Current literature on isoform-specific roles of AKT in cancer progression is full of contradictions. Despite lack of total consensus, literature favors the possibility that AKT1 is involved in increased proliferation and tumor growth as well as decreased apoptosis, whereas AKT2 is associated with increased migration, invasion, and metastasis. AKT3 appears to play a role in increasing both proliferation and metastasis [37]. Inhibition of AKT1 in MMTV-ErbB2/neu and MMTV-PyMT-induced mouse mammary tumors results in diminished tumor development due to lower expression of Ki-67 and cyclin D and increased apoptosis [38]. The protumorigenic role of AKT1 is evident from a study on miR-409-3p. miR-409-3p reduced proliferation, decreased invasion and migration of breast cancer cells in vitro by downregulating AKT1 [39]. Few studies have described the tumor suppressor role of AKT1 in breast cancer. For example, reduced activity of AKT1 has been associated with a dysregulation of p53 and DNA-damage induced transcription [40]. In another study, AKT1 was observed to be central to the reduction of breast cancer invasiveness by another tumor suppressor called TIS21. Specifically, TIS21 impacts motility and metastasis by reducing the assembly of the cytoskeleton. This TIS21-mediated decrease in cancer cell motility involves AKT1-dependent downregulation of diaphanous-related formin and decreased NOX4-mediated ROS formation [41]. Additional support for antimetastatic activity of AKT1 came from studies that examined the role of CXCR2 in metastasis. CXCR2-mediated breast cancer metastasis corelated with lower AKT1 expression [42]. Interestingly, we reported distinct prognostic significance of AKT in breast cancer based on subcellular localization. Nuclear localization of activated AKT (pS473) is associated with better prognosis [43]. Few of the discrepancies noted in the literature could, therefore, be due to lack of consideration to subcellular distribution of phosphorylated AKT in experimental models. 

Like AKT1, the role of the AKT2 isoform in breast cancer is complex and there are conflicting reports in the literature. Many studies have implicated AKT2 in proliferation and metastasis of various cancers. In a lung cancer cell line, for example, knockdown of AKT2 resulted in lower proliferation and invasiveness, which correlated with reduced retinoblastoma (RB) phosphorylation and COX2 expression [44]. In PTEN-deficient prostate tumors, AKT2 is necessary for growth and survival [45]. In breast cancer, AKT2 may increase metastatic potential via several mechanisms. For example, upregulation of AKT2 causes β1-integrin-mediated increase in adhesion and invasion via collagen IV. In this regard, AKT2 was found to localize specifically to collagen IV matrix during cell attachment [46]. Furthermore, AKT2 increases expression of the actin-bundling protein, palladin, which is associated with invasive breast cancer [47]. These reports collectively provide evidence for the role of AKT2 in breast cancer metastasis. There is also evidence for tumor suppressor function of AKT2. For example, AKT2 ablation was shown to result in an acceleration of tumor induction in MMTV-ErbB2/neu and MMTV-PyMT transgenic mice [38]. A consensus can be drawn in that while AKT1-mediated signals are associated with cell proliferation and survival, AKT2-mediated signals are associated with metastatic progression with limited or growth inhibitory actions on the primary tumor.

AKT3 has also been linked to breast cancer proliferation and survival. AKT3 is expressed in animal models of ErbB2+ tumors and contributes to proliferation [48]. Interestingly, there are reports that AKT3 is of a particular significance in Triple Negative Breast Cancer (TNBC). *AKT3* amplification or translocation with *MAGI3* gene, leading to constitutive AKT3 activity is reported in breast cancer [49]. Another study has shown that inhibition of AKT3 but not AKT1/2 leads to decreased mammosphere formation [50]. Predictably, AKT3 has been identified as a potential therapeutic target for the treatment of cancer. Treatment with miRNA-29b, which targets AKT3, caused reduced vascularization via modulation of VEGF and c-Myc levels, and reduced growth in vivo [51]. Furthermore, overexpression of AKT3 leads to lower expression levels of p53, p21, and p27, and increased expression of cyclin-D1, Bcl2, and XIAP [51]. 

*PIK3CA* and *AKT* isoforms are frequent targets of mutations/amplification in breast cancer. Almost 50% of breast cancers show genomic aberrations associated with these genes (Figure 2). It is interesting that at least 50% of tumors with *ESR1* mutation/amplification contain mutations/amplifications in the components of PI3K/AKT/mTOR pathway. 

## 4. Anti-Estrogen Therapies for Breast Cancer

Clinical, pathologic, and genomic scoring parameters determine whether anti-estrogen therapy is advisable either as a single agent or in combination with chemotherapies. Clinical and pathologic features include ERα and progesterone receptor positivity and involvement of lymph nodes. The genomic score includes the 21-gene recurrence score [53]. Anti-estrogen therapy involves small molecules that induce conformational changes in ERα that prevent E2 binding, cause ERα degradation, or block E2 synthesis by inhibiting the activity of aromatase. Based on these mechanisms of action, anti-estrogen therapies can be broadly divided into three groups: selective estrogen receptor modulators (SERMs), selective estrogen receptor down regulators/degraders (SERDs), and aromatase inhibitors (AIs) [54,55].

*SERMs*: SERMs are the most preferred type of treatment for ERα-positive breast cancer and they act by binding to ERα and suppress E2-regulated gene expression by enhancing corepressor instead of coactivator recruitment to ERα [56]. Examples of SERMs include tamoxifen, raloxifene, lasofoxifene, arzoxifene, bazedoxifene, toremifene, acolbifene, and ospemifene [54,57,58,59,60,61]. Tamoxifen is the most frequently used SERM to treat breast cancer. A summary of SERMs approved for clinical use in the treatment of hormone receptor positive breast cancer can be found in Table 1.

*SERDs*: SERDs are known as pure ER antagonists. Binding of SERDs to ERα disrupts dimerization, DNA binding, and aids premature proteosomal degradation of the receptor [64]. Fulvestrant is the only SERD currently approved for clinical use and can be a choice either in first line hormone therapy setting or after tamoxifen and AI failure [73,74]. SERDs currently in clinical development are summarized in Table 2.

*AIs*: Aromatase, encoded by *CYP19A1* gene, is an enzyme of cytochrome P450 family which is involved in biosynthesis of estrogens from androgen precursors. Aromatase is expressed in several estrogen-producing tissues including ovaries, breast, placenta, adrenal glands, testicles, adipose tissue, bone, liver, muscles, and brain. Systemic inhibition of estrogen biosynthesis by aromatase inhibitors (AIs) block ER signaling and consequently reduces circulating estrogen levels by more than 90% [71]. AIs are typically given under post-menopausal setting as aromatization of androgens is the main source of E2 at this stage. The long-term deprivation of estrogen, however, causes osteoporosis and hypersensitivity to the low level of estrogen. AIs are normally classified into two subtypes according to their chemical structure: steroidal (type I inhibitors) that includes testolactone, exemestane, formestane, and nonsteroidal (type II inhibitors) that includes letrozole, anastrozole, aminoglutethimide, and fadrozole [91,92]. Steroidal AIs, also known as suicidal inhibitors, first bind to the natural substrate binding site of the aromatase and become a reactive intermediate that covalently binds to aromatase resulting in irreversible inhibition. By contrast, nonsteroidal AIs bind noncovalently to heme moiety of aromatase and saturate its active site, thus, resulting in reversible inhibition. Breast cancers can acquire resistance to AIs after prolonged suppression of estrogen production by mechanisms other than those caused by fulvestrant or tamoxifen (described below). For this reason, tumors that have acquired resistance to AIs respond to other anti-estrogen therapies. Development of resistance to AIs is a major clinical concern in breast cancer and is an area of great research focus [93].

## 5. Mechanisms of Resistance to Anti-Estrogens

### 5.1. Ligand-Independent Activation of ERα

Sluyser and Mester (1985) proposed that mutations in ERα lead to ligand independent activation and mutated receptor may deregulate cell proliferation [94]. Earlier sequencing studies of primary and metastatic tumors with and without tamoxifen treatment revealed low frequency *ESR1* mutations [95]. Similar results were observed when *ESR1* was sequenced in ERα+ and ERα- tumors. Interestingly, about 1% of mutation frequency was observed in these early studies. In 1997, another group identified three missense mutations (Ser47Thr, Lys531Glu, and Tyr537Asn) in the *ESR1* gene [96] and the resulting mutant proteins displayed hyperactivity in the absence of ligand [97]. Later, a clinical sequencing program confirmed earlier published results and added new point mutations (Leu536Gln, Tyr537Ser, Tyr537Cys, Tyr537Asn, and Asp538Gly) to the list [98]. Interestingly, these mutations were acquired upon anti-estrogen treatment, observed mostly in metastatic tumors, showed constitutive activity at variable magnitude, and responded differentially to the SERD fulvestrant [75]. Another study with 625 postmenopausal and 328 premenopausal ERα+ tumors revealed that the ERα+ tumors are highly heterogenous and concluded that more comprehensive studies are required to explore whether *ESR1* mutations occur in primary tumors [99].

The second mechanism of ligand-independent activation involves receptor phosphorylation. At least 16 different amino acids in ERα have been suggested to undergo phosphorylation [100]. These residues include S46/47, Y52, S102/4/6, S118, S154, S167, S212, Y219, S236, S282, S294, S305, T311, Y537, S554, and S559. Interestingly, few of the *ESR1* mutations observed in breast cancer metastatic samples correspond to phosphorylatable residues (Y537, for example) suggesting relevance of these phosphorylations in ERα function. Kinases involved in these phosphorylations include PKC, c-Abl, GSK-3, ERK1/2, CDK2, CDK7, IKKα, mTOR/p70S6K, p90RSK, AKT, CK2, and SRC. 

Other post-translational modifications (PTMs) can influence ERα activity and potentially impact response to anti-estrogens. For example, PRMT1 methylates Arginine 260 within ERα DBD. This methylation is required for interaction with PI3K and SRC [101]. Furthermore, p300 acetylates ERα at Lysine 266 and 268 [102]. These acetylations enhance DNA binding and transactivation function of the receptor. Other modifications such as ubiquitination, SUMOylation, and palmitoylation have been shown to affect ERα stability, function, and localization [103]. Therefore, these PTMs can be potential prognostic or predictive biomarkers for tumor evaluation and response to anti-estrogens [103]. 

In addition to being potential prognostic biomarkers, components of the ubiquitin-proteasome system (UPS) have been suggested to be potential candidates for targeted therapies against the ER [104]. This is due in part to previously identified associations between functions of these components and ER expression or activity. For example, inhibition of polyubiquitination of ERα leads to an increase in the stability of the receptor [105]. Moreover, a recent report describes the ability of cardiac glycosides Ouabain and Digoxin to degrade ERα, potentially via activation of the proteasomal system, with subsequent inhibition of estrogen signaling, cell cycle blockade and apoptosis of primary and metastatic breast cancer cells [106]. The complex interaction between ERs and the UPS has been reviewed elsewhere [104]. 

### 5.2. Interplay between PI3K/AKT and ERα Signaling to Overcome the Effects of Anti-Estrogens

Our lab considered a nexus between PI3K/AKT and ER signaling and the role of this axis in anti-estrogen resistance in late 1990s, even before genomic revolution revealing enrichment of genomic aberrations of PI3K/AKT pathway genes in ERα+ breast cancers. Scientific premise for studies was based on the presence of consensus sequence (R-X-R-X-X-S/T) for AKT phosphorylation in ERα surrounding the amino acid S167 (RERLAST) [107]. Other groups subsequently reproduced data identifying the crosstalk between ERα and AKT signaling [108,109]. A cBioPortal [52] analysis shows ≈50% of breast cancers with genomic aberrations in *PIK3CA*, *AKT1*, *AKT2*, *AKT3*, and/or *ESR1*,suggesting relevance of this signaling axis in breast cancer (Figure 2). Below, we summarize our studies describing specific effects of AKT on ERα signaling and complement our studies with other reports in the literature. 

### 5.3. AKT Influences Genome-Wide Binding of ERα and E2-Mediated Gene Expression

The post-genomic era witnessed significant advances in our understanding of transcription initiation process, particularly binding of transcription factors to chromatin. Various groups used chromatin immunoprecipitation assay followed by microarray hybridization (ChIP-on-Chip) or sequencing (ChIP-seq) to map binding patterns of ERα to chromatin with and without E2 treatment [110,111]. Depending on the study, >3000 ERα binding regions, many of them enriched for EREs, were observed in E2-treated ERα+ cell line MCF-7. These types of studies also revealed the role of pioneer factors such as FOXA1 in binding of ERα to the genome. We used the ChIP-on-Chip assay of parental MCF-7 cells and MCF-7 cells overexpressing constitutively active AKT to determine the influence of AKT on genome wide DNA binding of ERα in vivo [112]. We coupled ChIP-on-Chip data with RNA microarray to correlate ERα binding to the genome with gene expression changes in E2 ± constitutively active AKT-dependent manner. We observed ≈40% changes in ERα binding patterns in cells with constitutively active AKT compared to parental cells and AKT caused an increase in the expression of E2-regulated genes that are enriched for the TGF-β, NF-κB/TNF, retinoic acid, and E2F pathways. Consequently, the AKT-overexpressing MCF-7 cells were resistant to TGF-β-induced growth inhibition compared to the parental MCF-7 cells. Furthermore, we reported a secondary role for overexpressed AKT that involved changes in the E2-regulated expression of E2F2 and E2F6 and secondary E2-response. AKT also altered E2-regulated expression of both oncogenic and tumor-suppressor microRNAs [113]. In a subsequent study, following the observation of a differential role of AKT1 and AKT2 in E2-regulated gene expression and the absence of an effect of individual AKT isoforms on E2 response in BT-474 cells, we postulated that the effects of PI3K/AKT signaling on the genomic activity of the ERα is dependent on cell type [114]. 

Further complexity in AKT-ERα crosstalk emerged during studies related to understanding the mechanisms of resistance to PI3K/AKT inhibitors. Toska and colleagues [115] observed that KMT2D, a histone methyltransferase, is central to activation of ERα by PI3K/AKT signaling. Inhibition of PI3K activity caused an impairment in AKT-mediated phosphorylation and subsequent inactivation of KMT2D. In cells treated with PI3K/AKT inhibitors, a compensatory pathway activated ERα through unphosphorylated KMT2D. KMT2D opened chromatin state at ERα binding sites that allowed recruitment of pioneer factors like FOXA1 and PBX1 and ERα-mediated transcription (Figure 3). Further studies identified another negative feedback system that involved SGK1. Authors observed that PI3K inhibition, which induced KMT2D activity with enhanced ERα transcriptional activity, also led to increased expression of SGK1. SGK1 subsequently phosphorylated KMT2D and impaired the ability of KMT2D to stimulate the transcriptional activity of the ERα in a negative feedback mechanism [116]. However, it is unknown whether genome wide binding patterns and transcriptional targets of ERα differ when it is activated directly through phosphorylation by AKT or following access to the genome under conditions with elevated KMT2D activity but lower AKT activity (Figure 3). Nonetheless, these results provide an explanation as to why PI3K/AKT-mediated resistance to anti-estrogens cannot be therapeutically overcome with PI3K/AKT inhibitors alone. A combination of PI3K/AKT inhibitors and SERDs may be required at the very least to block crosstalk between PI3K/AKT and ERα:E2 signaling.

### 5.4. ERα-Mediated Alternative Splicing and Influence of AKT

Alternative splicing is important for generation of complex and diverse proteomes that mediate cellular processes such as apoptosis, growth, motility, differentiation, and stem cell maintenance in response to various extracellular factors [117,118,119]. Genomic alterations that impact few of these cellular processes contribute to etiology and progression of cancer [120]. E2, via ERs, promotes alternative splicing of specific genes that affect breast cancer cell behavior. We demonstrated that AKT alters E2-mediated splicing of genes [121]. To identify endogenous targets of E2-ERα mediated alternative splicing and potential roles of AKT in splicing, we had previously utilized exon-specific microarray technique to evaluate patterns of alternative splicing in parental and AKT-overexpressing MCF-7 cells, with or without E2 treatment. This was preceded by a CD44 minigene splicing experiment that indicated a significant effect of AKT on E2-mediated alternative splicing. AKT specifically altered E2-mediated splicing of FAS/CD95, FGFR2 and AXIN-1 genes with consequent effects on FAS-mediated apoptosis and response to keratinocyte growth factor (KGF), a FGFR2 ligand [121]. Since FGFR2-mediated signaling counteracts the effects of tamoxifen [122], AKT-mediated resistance to anti-estrogens could involve its effects on E2-mediated FGFR2 splicing/signaling. 

### 5.5. AKT Is a Bridge between Growth Factor and ERα Signaling

Switching of ERα+ cancer cells from dependency on E2-mediated proliferative signals to growth factor-dependent signals is a major mechanism of resistance to anti-estrogens [123]. Due to intratumor heterogeneity, it is difficult to exclude the possibility that anti-estrogen resistance in some cases is due to clonal selection of de novo anti-estrogen resistant ERα+ cancer cells with inherently enhanced growth factor signaling capacity. In either way, the PI3K/AKT signaling axis, which is downstream of multiple extracellular growth factors including epidermal growth factor (EGF), platelet-derived growth factor (PDGF), and insulin-like growth factor, could serve as bridge between growth factors and ERα [124,125,126,127]. This interaction between ERα and growth factor signaling pathways impacts transcriptional activity of the ERα, both in the presence and absence of E2, considerably affecting response to breast cancer therapy. For example, Lupien and colleagues [128] showed the EGF can induce genome-wide binding of ERα and the genomic targets of ERα following induction by EGF are distinct from E2-induced genomic targets. These EGF-induced genomic targets of ERα overlapped with genes overexpressed in HER2-positive breast cancers. We propose that AKT is one of the mediators EGF-dependent ERα binding to the genome. Since EGF mediated genome-wide binding of ERα is independent of E2, EGF-induced genome-wide binding of ERα cannot be restrained by tamoxifen or AIs. Consistent with these findings, increased levels of pAKT and AKT kinase activity was observed in four out of six hormone resistant cell lines, with a concordant increase in sensitivity of the cell lines to hormonal therapy following inhibition of AKT phosphorylation by PI3K and AKT inhibitors [129]. Moreover, a retrospective study by Bostner et al. [108] showed an association between the activity of phosphorylated PI3K, AKT, and mTOR and resistance to tamoxifen therapy. 

Few studies have proposed an alternative mechanism to PI3K/AKT-mediated resistance to anti-estrogens. For example, an inverse correlation between PI3K activation scores and ER expression levels in ERα+ breast cancer has been described [130]. Increased PI3K activity was reported to be associated with a decrease in ERα expression and a concurrent development of resistance to hormonal therapy. This was observed through an analysis of proteomic and transcriptomic signatures of PI3K in ERα+ Luminal B breast tumors [130]. This report is consistent with a previous molecular-pathology study which revealed that loss of PTEN activity is associated with a decrease in ERα and progesterone receptor (PR) expression [131], a phenomenon that is likely to be due to uncontrolled PI3K activation and subsequent induction of AKT activity. Taken together, these reports provide evidence for an alternative mechanism by which PI3K and AKT wean cancer cells away from E2–ERα and alter cancer cell properties with consequent effects on their response to hormonal therapy. Therefore, the PI3K/AKT signaling axis is being evaluated as a probable target for the mitigation of resistance to endocrine therapy in breast cancer cells [13,132]. However, the potential for hyperactivity of feedback loops of the PI3K/AKT signaling cascade upon targeting this axis confounds the effectiveness of PI3K/AKT inhibitors as evident in clinical experiences described below [109,133,134]. 

## 6. Current Clinical Strategies to Treat Anti-Estrogen Resistant Breast Cancers

Many years of preclinical and translational research has enabled cataloging of the following signaling axis in resistance to endocrine therapies: genomic abnormalities in *ESR1*, CCND1-CDK4/6-RB, and PI3K-AKT-mTOR signaling pathways. 

### 6.1. Targeting Mutant ERα through New Class of SERDs

*ESR1* mutations are commonly acquired as a result of selective pressure of endocrine therapy that forces ERα to acquire ligand-independent signaling capabilities [135,136]. Inhibition of activities of these mutants by rationally designed novel therapeutic strategies has the potential to substantially improve outcomes. Fulvestrant has shown some efficacy in *ESR1*-altered ERα-positive breast cancers previously treated with SERMs and continues to be the treatment of choice at present [75]. However, there are other SERDs under development, which can degrade both wild type and mutant ERα to a similar degree. These include GDC0927, AZD9496, and RAD1901, which are in phase I, and GDC0810, which is in phase II clinical trials [76] (Table 1).

### 6.2. Inhibition of CCND1-CDK4/6-RB Pathway

The CCND1-CDK4/6-RB pathway is involved in cell cycle progression [137]. This pathway controls whether a cell arrests or advances at G1-S phase of the cell cycle. At this checkpoint, cyclin-D binds with CDK4/6 to promote progression of cell cycle via inhibition of tumor suppressor retinoblastoma (RB) protein. Approximately 35% of ERα+ breast cancers demonstrated amplification of *CCND1* gene (encoding cyclin-D1), and about 16% demonstrated amplification of the gene that encodes *CDK4* [138,139]. Moreover, loss of endogenous negative regulators of CDK4/6, *CDKN2A*,and *CDKN2C*, results in hyperactivity of CDK4/6 in ERα+ breast cancers [140]. This suggests the therapeutic utility of chemical inhibitors of CDK4/6 in ERα+ breast cancers [140,141]. The CDK4/6 inhibitors, palbociclib (PD-0332991), ribociclib, and abemaciclib, in combination with endocrine therapy are frequently used to treat recurrent ERα+ breast cancer and have improved progression free survival (PFS) [142,143]. In the PALOMA trial, combination of CDK4/6 inhibitors with the aromatase inhibitor letrozole exhibited improved PFS compared to letrozole alone [144]. However, the findings from PALOMA-1 trial demonstrated that genetic aberrations of *CCND1–CDK4/6* axis are not predictive for clinical efficacy of palbociclib treatment [145]. In the phase 3 MONALEESA-2 trial, ribociclib plus letrozole or tamoxifen significantly prolonged PFS in postmenopausal ERα+ breast cancers previously untreated with systemic therapy [146,147]. In the MONARCH-3 trial, abemaciclib with a nonsteroidal AI was used in postmenopausal ERα+ breast cancer and this treatment regime was associated with significantly increased median PFS [148]. CDK4/6 inhibitors with fulvestrant were approved by FDA as a line of treatment for endocrine therapy-resistant metastatic disease. Although CDK4/6 inhibitors are effective in improving PFS, eventual resistance to these inhibitors is an issue. Resistance mechanisms to these inhibitors are one of the intensely explored current research topics [137]. A summary of approved combination therapies including inhibitors of the CCND1-CDK4/6-RB pathway and those in clinical trials can be found in Table 2.

### 6.3. Inhibition of PI3K-AKT-mTOR Pathway

As noted above, mutations in PI3K-AKT-mTOR pathway genes are frequently observed in ERα+ breast cancers and at least 50% of breast cancers with *ESR1* mutation/amplifications displayed genomic aberrations of this pathway (Figure 2). Based on these observations as well as significant amount of preclinical data described above, it is logical to conduct clinical studies combining inhibitors of PI3K-AKT-mTOR pathways with anti-estrogens. Indeed, targeting the PI3K-AKT-mTOR pathway has been demonstrated to be beneficial in both neoadjuvant and advanced settings in ERα+ breast cancers [149,150]. Everolimus (Afinitor), a mTORC1 inhibitor, is a frontline drug that interrupts the PI3K-mediated signaling. It has been approved in combination with hormonal therapies to treat advanced postmenopausal ERα+ breast cancer [149]. In the BOLERO-2 trial, combination of everolimus with exemestane showed improved median PFS of 10.6 months; however, tumors with *PIK3CA* mutations were not responsive [87,127,149]. In the same trial, addition of everolimus to standard endocrine therapy demonstrated a potential predictive efficacy in patients with circulating *ESR1* mutations [151,152].

Several other PI3K-AKT pathway targeted therapies have been examined clinically. Tumors with *PIK3CA* mutations, which progressed after treatment with AI, showed improved PFS when treated with PIK3CAα isoform specific inhibitor alpelisib and fulvestrant combination [86]. The alpelisib plus letrozole combination revealed a clinical benefit along with higher tolerable toxicity profile [153]. However, the NEO-ORB trial that was carried out to evaluate the efficacy of the Letrozole-Alpelisib combination on response rate in the neoadjuvant setting showed no significant results [154]. Several phase II and III trials are still in progress with PIK3CAα-specific inhibitor to further determine the predictive therapeutic target value of *PIK3CA* mutations [155,156]. In earlier trials, combination of fulvestrant with pan-isoform PI3K inhibitors, i.e., buparlisib and pictilisib or β isoform-sparing PI3K inhibitor taselisib were evaluated, which indicated limited clinical benefits [157,158]. In the phase III BELLE-2 trial, initial results showed that patients with circulating DNA with *PIK3CA* mutations benefited from the combined treatment of PI3K inhibitor BKM120 and fulvestrant [159]. Despite some clinical efficacy, this combination therapy is not being pursued further due to toxicity profile.

Breast cancers with AKT mutation responded well to an ATP-competitive inhibitor Ipatasertib (GDC-0068) [160]. The pan-AKT inhibitors such as AZD5363 (Capivasertib), MK-2206, and GSK2141795 have been tested clinically but with limited benefits. The AKT1/2-inhibitor demonstrated good responses in preclinical studies, but it exhibited toxicity in clinical trials [161,162,163]. The addition of MK-2206 to anastrozole did not demonstrate a significant benefit to ERα+ breast cancer patients with *PIK3CA* mutations [164]. A summary of approved combination therapies including PI3K-AKT-mTOR pathway and those in clinical trials can be found in Table 2.

## 7. Conclusions and Future Directions

Intense preclinical research on nexus between ERα-E2 and PI3K-AKT-mTOR pathway has provided tangible benefits in clinical settings through effective combination therapies. The PIK3CAα-specific inhibitor alpelisib and the AKT inhibitor Ipatasertib are the two major success stories. Future research needs to focus on developing biomarkers that can predict response to such treatment as a first line therapy, developing combination therapies that are uniquely effective against *ESR1* mutated tumors as well as tumors with mutations in additional components of the PI3K-AKT-mTOR pathway. Considerable attention has to be given to feedback regulation in this pathway as well as “whack-a-mole” effects to derive effective combination therapies. Based on existing knowledge, it appears that PI3K-AKT-mTOR pathway inhibition is primed for such a “whack-a-mole” effect. PI3K-AKT-mTOR pathway is the major component of insulin signaling and it is natural for such a physiologically relevant pathway to have various feedforward and feedback loops to maintain homeostasis. Since ERα+ breast cancers tend to recur even after 20 years of initial diagnosis, new treatment strategies need to consider keeping residual tumor cells dormant forever or effectively eliminate dormant cells. One possible way to achieve this is further development of SERDs that are effective in degrading both wild type and mutant ERα with limited toxicity and can be administered in a cost-effective manner. Although ERα+ breast cancers harbor lower mutation load than TNBCs [140], heterogeneity due to acquired plasticity of cancer cells remains a major mechanism of resistance to targeted therapies and a better understanding of this plasticity will aid in the development of new therapies.

## Figures and Tables

**Figure 1 cancers-13-00369-f001:**
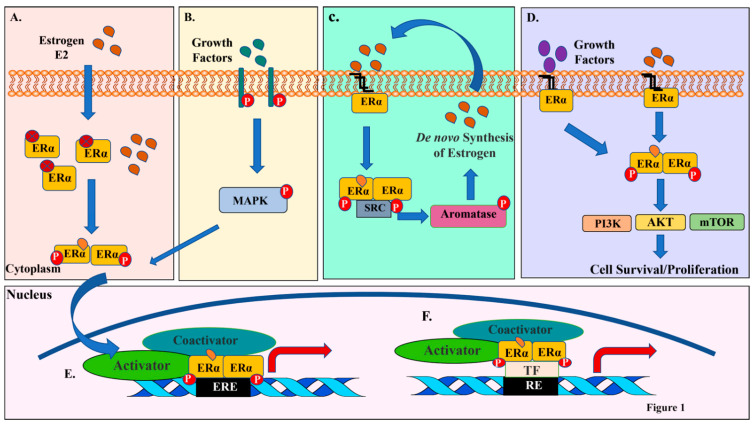
ERα:E2 signaling network and crosstalk with growth factor signaling. (**A**) Schematic view of classic genomic ERα:E2 signaling involving passive diffusion of E2 into cytoplasm and nucleus and activation of ERα signaling. (**B**) Growth factors activate receptor tyrosine kinases, which activate MAPKs. MAPKs can then phosphorylate and activate ERα either independent of E2 or synergize with E2 for optimal ERα activation. (**C**) Membrane associated ERα interacts and activates SRC kinase upon ligand binding. SRC kinase then phosphorylates and activates aromatase, which catalyzes conversion of androgens to estrogens within cells and amplify both genomic and nongenomic ERα-E2 signaling. (**D**) Membrane anchored ERα can also activate various cytoplasmic kinases including PI3K-AKT-mTOR pathway through nongenomic actions and these actions occur rapidly (within five minutes) after encountering the ligand. Activated PI3K-AKT-mTOR can enhance genomic actions of ERα (depicted in Figure 3). (**E**,**F**) Activated ERα alters gene expression in the nucleus through either direct binding to estrogen response elements (EREs) in the genome or bind genome by tethering onto other transcription factors (TFs) and their response element (RE).

**Figure 2 cancers-13-00369-f002:**
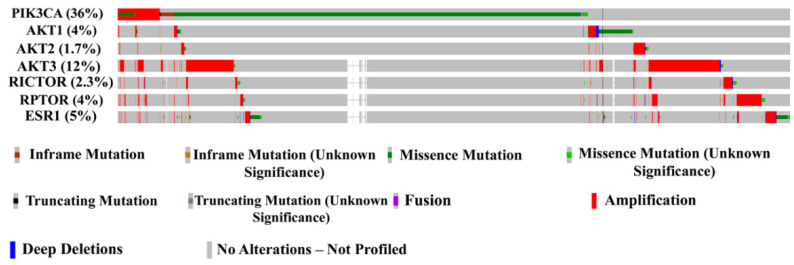
Co-occurrence of genomic aberrations in *PIK3CA*, *AKT1*, *AKT2*, *AKT3*, *ESR1*, and mTOR components *RICTOR* and *RPTOR* genes in breast cancer. cBioPortal database was used to create this figure [52].

**Figure 3 cancers-13-00369-f003:**
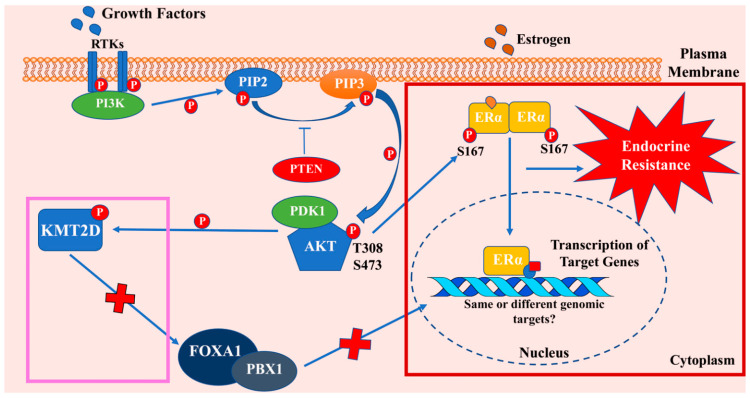
Nexus between PI3K/AKT and ERα pathways. Nexus between these two pathways involves both positive (red rectangle) and negative (pink rectangle) loops. Activation of PI3K leads to activation of AKT, which can directly phosphorylate ERα to promote ligand-independent activity and endocrine therapy resistance. However, AKT-mediated phosphorylation triggers degradation of the histone methyltransferase KMT2D, which is required for opening the chromatin region for binding of pioneer factors FOXA1 and PBX1. FOXA1 and PBX1 facilitate ERα binding to the genome. It is unknown whether genome wide binding and gene expression changes by ERα phosphorylated by AKT and that facilitated by KMT2D are different and contribute differentially to antiestrogen response.

**Table 1 cancers-13-00369-t001:** SERMs/SERDs/AIs that are clinically used or under clinical development.

Class of Drug	Mechanism of Action	Drugs Approved for Clinical Use	Drugs in Clinical Development
Selective estrogen receptor modulators (SERMs)	Suppression of E2-regulated gene expression by enhancing corepressor recruitment to ERα [56]	Tamoxifen, Toremifene, Raloxifene [62]	Bazedoxifene [62,63]
Selective estrogen receptor downregulators (SERDs)	SERDs disrupt ER dimerization and DNA binding and aid premature proteosomal degradation of the receptor [64]	Fulvestrant [57,65,66]	Elacestrant (RAD1901), AZD-9496, GDC-0927, LSZ102, SAR439859, G1T48 [66,67,68,69,70]
Aromatase inhibitors (AIs)	AIs prevent aromatase-mediated synthesis of estrogens from androgens. Thereby, decreasing circulating estrogen levels [71]	Exemestane (steroidal), Letrozole, Anastrozole (nonsteroidal) [72]	

**Table 2 cancers-13-00369-t002:** Combination therapies for anti-estrogen resistant breast cancers.

Current Strategies	FDA Approved Drugs/Drug Combinations and Associated Clinical Trials	Drugs/Drug Combinations in Clinical Trials
Target mutant ERα through new class of SERDs	Fulvestrant [75]	AZD9496, GDC0927, RAD1901, GDC0810 [76,77]
Inhibition of CCND1-CDK4/6-RB pathway	Palbociclib and Letrozole combination (PALOMA-2 trial) [78,79] Palbociclib and Fulvestrant Combination (PALOMA-3 trial) [80,81] Abemaciclib as monotherapy (MONARCH-1 trial) [82] Abemaciclib in combination with Fulvestrant (MONARCH-2 trial) [83] Abemaciclib combined with non-steroidal aromatase inhibitors letrozole or amastrozole (MONARCH-3 trial) [84]	Ribociclib (LEE011) in combination with endocrine therapy (Tamoxifen and Goserelin or a nonsteroidal aromatase inhibitor and Goserelin) (MONALEESA-7 trial) [85]
Inhibition of the PI3K-AKT-mTOR pathway	Alpelisib and Fulvestrant combination (SOLAR-1 trial) [86] Everolimus in combination with Exemestane (BOLERO-2 trial) [87,88]	Ipatasertib in combination with endocrine therapy and a CDK4/6 inhibitor (TAKTIC trial) [89]
Concurrent inhibition of ERα, CCND1-CDK4/6-RB pathway and the PI3K-AKT-mTOR pathways		Triplet therapy combining Palbociclib, Taselisib and Fulvestrant and doublet therapy combining Palbociclib and Taselisib [90]

## Data Availability

The data presented in this study are openly available in cBioPortal (ref. [52]).

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
