# Peer review of "Nexus between PI3K/AKT and Estrogen Receptor Signaling in Breast Cancer"

_cancers, 2021, doi:10.3390/cancers13030369_

Round 1

Reviewer 1 Report

Nakshatri and colleagues discussed the interaction between ER and PI3K/AKT/mTOR pathways signaling with insight on mechanisms of anti-estrogens resistance. Authors summarized anti-estrogen therapies for breast cancer, PI3K-AKT-mTOR signaling axis, and recent clinical strategies to tackle anti-estrogen resistance with a focus on PI3K-AKT-mTOR signaling pathways. The review needs restructuring to focus on the conclusion:

comments below;

1) The general understanding of the PI3K-AKT-mTOR signaling axis in breast cancer (currently under Anti-estrogen therapies for breast cancer; should be section3 instead of section2) should come after the biology of ER (section 2).

2) The therapeutic expect of PI3K-AKT-mTOR signaling axis (discussion of the studies dealing with inhibitors) in breast cancer could stay with modified/appropriate heading in section3; Anti-estrogen therapies for breast cancer).

3) The discussed studies in sections anti-estrogen therapies for breast cancer and current clinical strategies to treat anti-estrogen in-line to support the nexus between PI3K-AKT-mTOR and Estrogen Receptor should also be presented as tables with references.

4) The abbreviations (e.g. ERE, RE) in figure1 should be spelled out in the legend.

5) The font size of texts in Figure 2 should be large enough to read in 100% view.

Author Response

Comments:

1) The general understanding of the PI3K-AKT-mTOR signaling axis in breast cancer (currently under Anti-estrogen therapies for breast cancer; should be section3 instead of section2) should come after the biology of ER (section 2).

Response: This section has now become section 3 and before section anti-estrogen targeted therapies

2) The therapeutic expect of PI3K-AKT-mTOR signaling axis (discussion of the studies dealing with inhibitors) in breast cancer could stay with modified/appropriate heading in section3; Antiestrogen therapies for breast cancer).

Response: We have retained this section as before

3) The discussed studies in sections anti-estrogen therapies for breast cancer and current clinical strategies to treat anti-estrogen in-line to support the nexus between PI3K-AKT-mTOR and Estrogen Receptor should also be presented as tables with references.

Response: We have included two tables in the revised manuscript with references

4) The abbreviations (e.g. ERE, RE) in figure1 should be spelled out in the legend.

Response: necessary corrections have been made.

5) The font size of texts in Figure 2 should be large enough to read in 100% view.

Response: This figure has been modified

Reviewer 2 Report

Aditi S Khatpe and co-authors described ER biology and ER-related signaling to study resistance mechanisms of ER+ breast cancer to endocrine therapy. In this connection, the authors paid a special attention to a nexus between PI3K/AKT and ER-signaling in breast cancer and presented very serious investigation. I value this work highly. At the same time, I have some critical remarks.

- In page 8, the authors write: “Other modifications such as ubiquitination, SUMOylation and palmitoylation have been shown to affect ER stability, function and localization. Therefore, these PTMs can be potential prognostic or predictive biomarkers for tumor evaluation and response to anti-estrogens”. However, the authors missed a large layer of research concerning the regulation of ER by ubiquitin proteasome system (doi:10.3390/biom10040500). Really, components of ubiquitin proteasome system can be not only potential prognostic or predictive biomarkers, but targets for anti-ER+ breast cancer therapy (doi:10.3390/cancers12123840). The authors should note it in their MS.

-  In Fig. 2, the in-figure letters are very small. They are legible only after high zoom. Therefore, their size should be increased.

I think the MS may be published after minor revision. 

Author Response

In page 8, the authors write: “Other modifications such as ubiquitination, SUMOylation and palmitoylation have been shown to affect ER stability, function and localization. Therefore, these PTMs can be potential prognostic or predictive biomarkers for tumor evaluation and response to anti-estrogens”. However, the authors missed a large layer of research concerning the regulation of ER by ubiquitin proteasome system (doi:10.3390/biom10040500). Really, components of ubiquitin proteasome system can be not only potential prognostic or predictive biomarkers, but targets for anti-ER+ breast cancer therapy (doi:10.3390/cancers12123840). The authors should note it in their MS.

Response: Thank you for your suggestions. These references have been included (Ref. 79 and 81) and discussed (lanes 365-376)

In Fig. 2, the in-figure letters are very small. They are legible only after high zoom. Therefore, their size should be increased.

Response: We have modified the figure and increased the size.